# Acute Postural Effects of Spinal Cord Injury: Dual Neural Opioid and Endocrine Non-Opioid Mechanism

**DOI:** 10.3390/cells14130980

**Published:** 2025-06-26

**Authors:** Hiroyuki Watanabe, Igor Lavrov, Mathias Hallberg, Jens Schouenborg, Mengliang Zhang, Georgy Bakalkin

**Affiliations:** 1Department of Pharmaceutical Biosciences, Biomedicinskt Centrum BMC, Uppsala University, Husargatan 3, 751 24 Uppsala, Sweden; nabe.1088640.ofq@gmail.com (H.W.); mathias.hallberg@uu.se (M.H.); 2Department of Neurology, Mayo Clinic, Rochester, MN 55905, USA; igor.lavrov@gmail.com; 3Neuronano Research Center, Department of Experimental Medical Science, Lund University, 221 84 Lund, Sweden; jens.schouenborg@med.lu.se; 4Department of Molecular Medicine, University of Southern Denmark, 5230 Odense, Denmark; mzhang@health.sdu.dk

**Keywords:** spinal cord injury, lateral hemisection, asymmetric postural deficits, opioid system, endocrine pathway, neurohormones

## Abstract

Lateral spinal cord injury including lateral hemisection (LHS) leads to asymmetric postural and motor deficits. After traumatic brain injury, asymmetric postural deficits are partly developed through activation of opioid receptors. We here characterized the effects of LHS on hindlimb postural asymmetry (HL-PA), a proxy for neurological impairments, and assessed the involvement of opioid system. In acute experiments on rats, high lumbar LHS induced HL-PA, characterized by ipsilateral hindlimb flexion. This asymmetry persisted after complete spinal cord transection at the hemisection level. Treatment with naloxone, a general opioid antagonist, abolished HL-PA both before and after transection, suggesting that the LHS effects are mediated through opioid receptors and that neuroplasticity of lumbar opioid circuits underlies the persistent asymmetry. Surprisingly, cervical LHS performed after complete lumbar spinal cord transection also led to HL-PA. However, the hindlimb was flexed on the contralateral side, and the effect was resistant to naloxone treatment. This asymmetry may be caused by endocrine factors, which convey side-specific messages through the humoral pathway after their release from supraspinal structures. Thus, after lateral spinal cord injury, the asymmetric postural deficits may be driven by an interplay between opposing lumbar opioid and neuroendocrine non-opioid mechanisms.

## 1. Introduction

Spinal cord injury (SCI) leads to postural and motor impairments that significantly affect quality of life. Lateral hemisection of the spinal cord (LHS) including Brown-Séquard Syndrome results in ipsilateral postural deficits and proprioceptive loss, along with contralateral deficits in pain and temperature sensation [1,2,3,4]. These impairments are caused by the disruption of descending motor pathways and may be exacerbated by neuroplasticity of spinal neurocircuits [5,6]. Local spinal networks and descending pathways are reorganized, proprioceptive afferents promote synaptic rewiring, and corticospinal and rubrospinal tracts form new connections to bypass injury sites [7,8]. Maladaptive plastic changes lead to hyperreflexia, spasticity, and spasms [5,6].

The endogenous opioid system in the spinal cord is involved in regulation of sensory and motor processes [9,10,11,12,13]. In the dorsal horn, it gates pain signals, whereas in the ventral spinal cord, opioid receptor activity may regulate motor circuits. After SCI, morphine acting through µ-opioid receptors impairs motor recovery [14,15], whereas the dynorphin-κ-opioid receptor signaling regulates scar formation [16]. Block of opioid receptors can counteract changes in posture induced by unilateral traumatic brain injury. For example, naloxone, a general opioid antagonist, abolished hindlimb postural asymmetry (HL-PA) induced by brain injury [17,18,19,20].

Injury to one side of the brain leads to motor and postural abnormalities mostly on the opposite side of the body. These contralateral effects have long been attributed to the decussation of descending motor pathways, a cornerstone of neurology. However, growing evidence suggests that endocrine signaling also contributes to such lateralized outcomes [17,19,20]. In rats with complete cervical or thoracic spinal cord transection—where descending neural input is eliminated—unilateral cortical injury induces HL-PA, typically manifesting as contralateral hindlimb flexion. This motor response is accompanied by lateralized changes in reflexes and gene expression in the lumbar spinal cord, implying involvement of humoral factors [17,20]. Supporting this notion, serum from brain-injured animals induces HL-PA in naïve recipients, while hypophysectomy abolishes the injury effect—implicating the hypothalamic–pituitary axis [17].

Neurohormones have been linked to side-specific effects: β-endorphin and arginine vasopressin are associated with right-side motor responses following left hemisphere injury, while dynorphin and Met-enkephalin may mediate left-sided responses after right-hemisphere lesions [20]. These findings point to a lateralized neuroendocrine axis—termed the topographic neuroendocrine system—capable of encoding injury laterality into neurohormones and relaying these signals via the circulation to peripheral tissues. The proposed topographic neuroendocrine system may operate through a three-step process: (i) encoding of lateralized neural activity into neurohormonal signals by hypothalamic and pituitary structures; (ii) systemic dissemination through the bloodstream; and (iii) decoding into side-specific physiological effects at peripheral targets such as the spinal cord [17,20].

Although previous work has focused on brain injury, we hypothesized that the LHS may similarly engage the topographic neuroendocrine system. Given its role as a neuroendocrine integrator, the hypothalamus could detect asymmetric spinal input—potentially ascending via neural pathways—and orchestrate the release of side-specific neurohormones. These hormones, once in circulation, may influence motor output via direct action or pituitary-mediated cascades.

In this study, we tested whether the postural consequences of LHS involve opioid mechanisms and whether humoral signaling contributes to HL-PA following right-sided SCI. Our results suggest that both the neural pathway, controlled by the opioid system, and endocrine signaling are involved, and that the topographic neuroendocrine system serves to translate lateralized spinal inputs into asymmetric hormonal signals that evoke motor responses.

## 2. Materials and Methods

### 2.1. Animals

Male Wistar rats (Janvier Labs, Le Genest-Saint-Isle, France) weighing 150–200 g were used in this study. The animals received food and water ad libitum and were kept in a 12 h day–night cycle (light on from 10:00 p.m. to 10:00 a.m.), at a constant environmental temperature of 21 °C (humidity: 65%), and randomly assigned to their respective experimental groups. Experiments were performed from 9:00 a.m. to 8:00 p.m. After the experiments were completed, the animals were given a lethal dose of pentobarbital or perfused with phosphate-buffered saline (PBS) with 4% paraformaldehyde.

Approval for animal experiments was obtained from the Malmö/Lund ethical committee on animal experiments (No. 15317-21), approved on 24 November 2021.

The rats were given either 5 mg/kg of naloxone (Sigma-Aldrich, St. Louis, MO, USA) or a saline control solution via intraperitoneal injection.

### 2.2. Lateral Hemisection of the Spinal Cord

The rats were anesthetized with 3% isoflurane (Abbott, Fornebu, Norway) in a mixture with 65% nitrous oxide and 35% oxygen. The core temperature of the animals was controlled using a feedback-regulated heating system. Anaesthetized animals were mounted onto the stereotaxic frame, and the skin of the back was incised along the midline at the level of the lower thoracic or cervical vertebrae. After the back muscles were retracted to the sides, a laminectomy was performed at the T8 and T9 vertebrae or the C6 and C7 vertebrae.

Under a surgical microscope, the spinal cord was hemisected on the right side using micro-scissors, following local lidocaine application at the L1–L2 or C6–C7 levels [21,22]. To ensure precision, a 30-gauge needle was vertically inserted through the spinal cord midline, with the beveled side oriented toward the right hemi-cord. The needle was advanced to penetrate the entire spinal cord and reach the ventral wall of the vertebral canal. One tip of the iridectomy/microsurgical scissors was inserted into the needle track at the midline, while the other tip was positioned along the lateral surface of the right hemi-cord. A complete hemisection of the right hemi-cord was then performed using the scissors. To confirm the completeness of the hemisection, the lateral edge of the needle was used as a knife to cut through the lesion gap. The procedure was further verified by visualizing the vertebral canal and ensuring a clear lesion gap under the surgical microscope. A piece of Spongostan (Medi-spon MDD) was placed between the rostral and caudal stumps of the spinal cord. The injury was verified during autopsy. After completion of all surgical procedures, the wounds were closed with 3-0 suture (AgnTho’s, Lidingö, Sweden), and the rat was kept under an infrared radiation lamp to maintain appropriate body temperature during monitoring of postural asymmetry.

For sham surgery, animals underwent the same anesthesia and surgical procedures, but the hemisection was not performed.

### 2.3. Spinal Cord Transection

In the first set of experiments, 4 h after the right hemisection at the L1–L2 level, a 3–4 mm spinal cord segment, including the hemisection, was dissected and removed [17,20]. In the second set of experiments, prior to the right-side hemisection at the C6–C7 level, a 3–4 mm spinal cord segment between the two vertebrae level was dissected at the L1–L2 and removed. The completeness of the transection was confirmed by (i) inspecting the cord during the operation to ensure that no spared fibers bridged the transection site and that the rostral and caudal stumps of the spinal cord were completely retracted; and (ii) examining the spinal cord in all animals after termination of the experiment.

### 2.4. Histological Analysis of SCI

To evaluate the extent and precision of the hemisection, spinal cords from three rats with right-lumbar LHS and three rats with right-cervical LHS were processed and analyzed using Nissl staining. Three hours after LHS, HL-PA was measured, and, without further spinal cord transection and pharmacological experiments, the rats were transcardially perfused with 200 mL of saline, followed by 200 mL of 4% paraformaldehyde in 0.1 M PBS (pH 7.4). Spinal cords were carefully extracted, post-fixed overnight in 4% paraformaldehyde, and then transferred to PBS containing 30% sucrose for cryoprotection. Samples were stored at 4 °C for up to one week. A 10 mm segment of the spinal cord, including the hemisection site, was sectioned coronally into 20 μm slices.

Five sets of consecutive sections were collected, with sets 1, 3, and 5 stained using 1% toluidine blue (Sigma-Aldrich, St. Louis, MO, USA) according to the manufacturer’s protocol. To determine the maximal lesion area, three adjacent sections corresponding to the submaximal lesion region were selected. The lesion areas were outlined on each section, and the total lesion area was calculated by stacking these outlines. The maximal lesion area was defined as the total area occupied by the lesion after stacking, expressed as a percentage of the total cross-sectional area.

### 2.5. Analysis of Hindlimb Postural Asymmetry (HL-PA) by the Hands-On and Hands-Off Methods

The HL-PA value and the side of the flexed limb were assessed as described elsewhere [17,20]. Briefly, the measurements were performed under isoflurane anesthesia. The level of anesthesia was characterized by a barely perceptible corneal reflex and a lack of overall muscle tone. The anesthetized rat was placed in the prone position on the 1 mm grid paper.

In the hands-on analysis, the hip and knee joints were straightened by gently pulling the hindlimbs backward for 1 cm to reach the same level. Then, the hindlimbs were set free, and the magnitude of postural asymmetry was measured, in millimeters, as the length of the projection of the line connecting symmetric hindlimb distal points (digits 2–4) on the longitudinal axis of the rat. The procedure was repeated six times in immediate succession.

In the hands-off method, silk threads were glued to the nails of the middle three toes of each hindlimb, and their other ends were tied to one of two hooks attached to the movable platform that was operated by a micromanipulator constructed in the laboratory (Figure 1) [17,20]. To reduce potential friction between the hindlimbs and the surface with changes in their position during stretching and after releasing them, the bench under the rat was covered with plastic sheet, and the movable platform was raised up to form a 10° angle between the threads and the bench surface. The limbs were adjusted to lie symmetrically, and stretching was performed over a distance of 1 cm at a rate of 2 cm/s.

The threads then were relaxed, the limbs were released, and the resulting HL-PA was visually assessed or photographed. The procedure was repeated six times in succession; the postural asymmetry size was measured, in millimeters, as the length of the projection of the line connecting symmetric hindlimb distal points (digits 2–4) on the longitudinal axis of the rat; and the HL-PA values were used in statistical analyses.

The limb that projected over a shorter distance from the trunk was considered to be flexed. The HL-PA was measured in mm with negative and positive HL-PA values that were assigned to rats with the left- and right-hindlimb flexion, respectively.

The HL-PA data obtained using the hand-on method and the unbiased hand-off method showed high, significant correlations (see Figure 1 and Figure 2). Other data presented in Figure 1 and Figure 2 are for the hands-off assay.

### 2.6. Statistical Analysis

Experimental data were processed and statistically analyzed after completion of the experiments by a statistician who was not involved in their execution and not associated with this study. No intermediate assessment was performed to avoid any bias in data acquisition. Experimenters were not blind because the signs of the asymmetry were evident after LHS, and the LHS animals were used for analysis of the antagonists (saline).

Statistical analyses included repeated-measures ANOVA with Tukey’s HSD post hoc tests for multiple comparisons, two-tailed Student’s *t*-tests, and Fisher’s Exact two-tailed test. A Kolmogorov–Smirnov test confirmed that data did not significantly deviate from a normal distribution.

### 2.7. Declaration of Generative AI and AI-Assisted Technologies in the Writing Process

During the preparation of this work, the author (G.B.) used ChatGPT4o in order to polish the text. After using this service, the author reviewed and edited the content as needed and takes full responsibility for the content of the publication.

## 3. Results

### 3.1. The Lumbar LHS-Induced HL-PA

HL-PA, a model for neurological deficits, provides a rapid, reliable measure of side-specific responses following unilateral neurotrauma [17,20]. This binary model, producing either left- or right-sided responses, replicates conditions such as hemiparesis and spastic dystonia secondary to traumatic brain injury and stroke.

The HL-PA method has been extensively validated in previous studies [17,20], including comparisons between hands-on and hands-off assessments of asymmetry, as well as with biomechanical analysis of hindlimb musculo-articular resistance to stretch. In the hands-off approach and biomechanical assessments, a micromanipulator-controlled force measurement system was employed. This setup, comprising two digital force gauges mounted on a movable platform, enabled precise and unbiased quantification of hindlimb asymmetry (Figure 1C). Strong and statistically significant correlations were observed among all three assessment methods, confirming the robustness and reproducibility of the HL-PA measurements. In the present study, both hands-on and hands-off methods were used and yielded highly correlated results for HL-PA magnitude (Figure 1 and Figure 2). All other reported data were derived from the objective hands-off assay. To eliminate bias, data processing and statistical analysis were conducted post-experimentation by a statistician who was blinded to the experimental conditions.

In the first set of experiments, a right-sided LHS was performed at the L1–L2 level (Figure 1A,B). Four hours post-hemisection, the spinal cord was completely transected by removing a 3–4 mm segment at the hemisection level. HL-PA measurements were taken at baseline, three hours post-hemisection, one-hour post-naloxone or saline administration, and one hour after complete transection. Analysis of the lesion site in three rats showed that the right side of the spinal cord was nearly completely severed, with minor involvement of the left side (Figure 1E). The maximal lesion areas measured 48.4%, 51.8%, and 55.8% of the total cross-sectional area, respectively.

Three hours after LHS, rats showed robust HL-PA, which was negligible in sham-operated controls (Figure 1F). Hindlimb was flexed on the ipsilesional side. Repeated-measures ANOVA revealed no significant main effect of treatment with naloxone (F(1,10) = 1.71; *p* < 0.22), but it showed a significant main effect of time/repeated measures (F(3,30) = 34.85, *p* < 1 × 10^−5^) and a significant treatment—time interaction (F(3,30) = 6.87, *p* < 0.001), suggesting time-dependent differences in treatment response. Tukey’s HSD post hoc tests for this interaction showed significant differences between HL-PA values measured 3 h post-hemisection and the baseline values (*p* < 0.001 for both the “saline” and “naloxone” groups). Furthermore, the HL-PA values in rats with hemisected spinal cords were significantly higher than in sham-operated rats (Student’s *t*-test; *p* < 9 × 10^−5^).

Following naloxone administration, HL-PA was significantly reduced by approximately 75% relative to pre-injection levels and the saline-treated group (Tukey’s HSD post hoc test; *p* < 0.001), suggesting that LHS effects are mediated by opioid receptor activation. No significant changes in HL-PA were observed after complete spinal cord transection, suggesting that HL-PA may persist due to lumbar spinal cord plasticity induced by LHS. The proportion of rats with right-sided flexion after right-sided hemisection (15 with right flexion and none with left flexion) differed significantly from a random 50% left/50% right distribution (Fisher’s Exact Test, two-tailed: *p* = 0.002). The range of differences between the maximum and minimum HL-PA values was greater after naloxone treatment than before treatment in the same rat group and after saline treatment in the respective control LHS rats. However, a two-tailed, two-sample, unequal-variance Student’s *t*-test revealed significant differences in these comparisons: *p* = 0.013 and 0.032, respectively. Furthermore, an F-test for equality of variances revealed no significant differences in standard deviations between time points before and after naloxone administration (*p* = 0.201) or between the naloxone- and saline-treated groups (*p* = 0.366).

### 3.2. The Cervical LHS-Induced HL-PA in Rats with Complete Transection of Lumbar Spinal Cords

Unilateral brain injury can induce HL-PA through humoral pathway in animals with transected spinal cords [17,20]. To test the involvement of a neuroendocrine mechanism in LHS effects, HL-PA was assessed in rats with right-sided cervical hemisection at C6-C7, performed 1 h after a complete transection of the spinal cord at L1–L2 (Figure 2). HL-PA was measured before and 3 h after hemisection. Naloxone was administered 2 h post-hemisection. The right side of the spinal cord was almost entirely severed, with minimal involvement of the left side (Figure 2D). The maximal lesion areas for the three rats were 41.7%, 45.7%, and 52.3% of the total cross-sectional area, respectively.

The HL-PA size was higher in rats with right-sided cervical LHS than in (i) the same rats 1 h after complete spinal transection and (ii) rats with transected spinal cords after sham LHS (Figure 2E). Cervical hemisection induced flexion of the left hindlimb. Repeated-measures ANOVA revealed a significant main effect of hemisection (F(2,32) = 21.79, *p* < 1 × 10^−5^) and a significant interaction effect (F(4,32) = 7.28, *p* < 3 × 10^−4^), indicating differential responses over time among treatment groups. However, the main group effect was not significant (F(2,16) = 0.64, *p* < 0.54). With respect to the “saline” and “naloxone” groups, post hoc tests identified significant HL-PA differences between the post-LHS time point and both the pre-hemisection and post-transection time points (see Figure 2E for *p*-values). In addition, post hoc comparisons confirmed a significant difference between the LHS groups and sham-surgery group. No significant differences were observed between the “naloxone” and “saline” groups at any time point.

The proportion of rats with left-sided flexion after right cervical LHS (13 with left flexion and none with right flexion) differed significantly from both (i) rats assessed one-hour post-transection (combined hemisection and sham-surgery groups: 10 with left flexion vs. 7 with right flexion) and (ii) a random 50% left/50% right flexion distribution (Fisher’s Exact Test, two-tailed: *p* = 0.01 and *p* = 0.001, respectively). The difference in the proportions of left and right flexions between the lumbar and cervical LHS groups was highly significant (Fisher’s Exact Test, two-tailed: *p* = 3 × 10^−8^).

## 4. Discussion

### 4.1. LHS-Induced HL-PA

The first finding of this study is that right-sided lumbar LHS induces HL-PA characterized by ipsilateral hindlimb flexion. This effect persisted after complete spinal transection caudal to, or at the level of, the hemisection—suggesting that the asymmetry arises from neuroplastic changes established in the lumbar spinal circuits prior to spinalization.

The emergence of ipsilateral flexion implies that asymmetric descending activity from the injury site induces plastic rearrangements in motor circuitry below the lesion. These findings align with earlier reports showing enhanced monosynaptic and polysynaptic reflex activity on the ipsilateral side following LHS, even after complete transection performed caudally [23,24].

As in previous HL-PA studies [17,18,19,20], no nociceptive stimuli were applied, and tactile input was minimal upon the asymmetry analysis. Previously, it has been well established that stretch and postural reflexes are abolished for days following complete spinal cord transection [25,26,27], and are significantly suppressed under anesthesia [28,29]. Our findings suggest that neither stretch reflexes nor nociceptive withdrawal responses contribute to HL-PA formation or maintenance in spinalized, anesthetized LHS rats. However, some proprioceptive circuits, particularly those involving group II muscle afferents, may remain active after acute spinalization and contribute to tone regulation [30,31,32]. Together, the data support the idea that HL-PA is a multifactorial phenomenon, potentially arising from (i) persistent asymmetric activity of lumbar motoneurons independent of afferent drive; and/or (ii) tonic activation of proprioceptive neurons—perhaps via group II afferents—that maintain baseline muscle tone.

Individuals with stroke or cerebral palsy often exhibit persistent muscle activation in the absence of voluntary effort—termed spastic dystonia, defined as “stretch- and effort-unrelated sustained involuntary muscle activity following central motor lesions” [33,34,35,36,37,38,39,40]. Its pathogenesis may involve central, reflex-independent mechanisms that may be similar with that of HL-PA induced by the LHS. In this context, HL-PA may serve as a rat model for studying this centrally driven, asymmetric motor phenomenon.

### 4.2. The Opioid HL-PA Mechanism

The neurotransmitter mechanisms of spinal neuroplasticity following unilateral neurotrauma remain unidentified, with the exception of a role of the opioid system [17,18,19,20]. Searches for neurotransmitters mediating HL-PA formation demonstrated that opioid peptides and synthetic opioids, along with Arg-vasopressin, may be involved—they induce HL-PA in spinalized animals [17,18,19,20]. These responses were side-specific: κ-opioid agonists such as dynorphin and bremazocine induced left-hindlimb flexion, whereas δ-opioid agonist Leu-enkephalin and Arg-vasopressin elicited right-hindlimb flexion. The fact that HL-PA can be elicited by intrathecal administration of opioid peptides and synthetic opioid agonists in spinalized animals demonstrates that these effects are mediated though spinal circuits.

Pharmacological studies with opioid antagonists have shown that the opioid system is essential for HL-PA induced by unilateral brain injury, including ablation injury and controlled cortical impact of the sensorimotor cortex [17,18,19,20]. Consistent with these reports, our current findings suggest that opioid signaling contributes to HL-PA following the right-sided lumbar LHS. The non-selective opioid antagonist naloxone abolished postural asymmetry both before and after complete spinal transection, indicating that opioid mechanisms at the spinal level encode the LHS-induced response. We propose that the balance between mirror-symmetric spinal circuits regulating left- and right-hindlimb musculature is maintained by endogenous opioid tone. After LHS, this equilibrium may become disrupted by a side-specific activation of spinal opioid receptors, leading to asymmetric motor output.

The side-specific effects of opioids suggest that opioid receptors are lateralized within the spinal cord, and that these asymmetrically distributed receptors may differentially regulate the mirror-symmetric spinal circuits controlling left- and right-hindlimb muscles. Previous work identified asymmetric expression of opioid receptor genes in the cervical spinal cord [19,41], with all three receptor types (μ, δ, and κ) showing left-side dominance. The relative proportions of these receptors differed between the left and right spinal halves, and expression patterns were coordinated between dorsal and ventral regions, though differently on each side. These findings were then extended to the lumbar spinal cord, where similar lateralization patterns were observed. δ-Opioid receptor (*Oprd1*) expression was enriched on the left side, whereas the κ/δ receptor ratio (*Oprk1*/*Oprd1*) was higher on the right. Opioid peptides were also lateralized: Leu-enkephalin-Arg (a prodynorphin marker) and the Leu-enkephalin-Arg-to-Met-enkephalin-Arg-Phe ratio were greater on the left, consistent with the elevated prodynorphin-to-proenkephalin mRNA ratio in this region. In contrast, Met-enkephalin-Arg-Phe (a proenkephalin marker) was enriched on the right side. These findings indicate that the lateralized organization of the spinal opioid system may provide a molecular substrate for side-specific modulation of neural activity in response to unilateral brain or spinal cord injury.

Opioid peptides and receptors are expressed in interneurons in the dorsal and ventral spinal cord, where they regulate processing of sensory information, reflexes, and motor functions [9,42,43,44]. Opioid receptors are expressed by V1 inhibitory interneurons, which include premotor Ia interneurons mediating inhibition of antagonist muscles, and Renshaw cells mediating motor neuron recurrent inhibition but not motoneurons [9]. Dynorphins are key components of a spinal inhibitory circuit and expressed by distinct subpopulations of inhibitory and excitatory neurons [42]. Opioids modify ventral root reflexes via presynaptic inhibition of afferent signaling, via postsynaptic repression of interneurons in the dorsal horn, and by acting on interneurons regulating the activity of motoneurons in the ventral horn afferents [11]. Notably, spinal motor actions of opioids and opioid peptides are selectively directed onto pathways from flexor reflex afferents.

The model depicted in Figure 3 illustrates the proposed mechanism by which the opioid system mediates the LHS-induced ipsilateral response. Under normal conditions, descending neural projections tonically inhibit opioid peptide-producing interneurons in the spinal cord (Figure 3A). Following LHS, the loss of the descending control leads to the release of enkephalins or dynorphins on the injured side. Opioid peptides act on a neuronal subpopulation, which negatively regulates hindlimb motoneurons (Figure 3B). Activation of opioid receptors on these interneurons suppresses their inhibitory activity, resulting in disinhibition of motoneurons. This causes pathological hindlimb responses, such as persistent muscle contraction (Figure 3B). The administration of naloxone reverses these effects by restoring the activity of the opioid-sensitive interneurons (Figure 3C). Consequently, motoneuron firing is suppressed, pathological responses on the ipsilesional side are reduced, and symmetry in posture and reflexes is restored. Plasticity of the opioid system may contribute to the maintenance of the pathological state after complete disconnection between the injury site and the lumbar spinal cord.

Our opioid-related findings are consistent with preclinical and clinical studies reporting that general opioid antagonists may reverse asymmetric neurological deficits following unilateral cerebral ischemia and progressive multiple sclerosis [45,46,47,48,49,50,51,52,53,54]. These observations, together with our data, underscore the importance of identifying pathophysiological and molecular signatures of asymmetric motor deficits—such as hemiparesis and hemiplegia—that may be selectively mediated by different opioid receptor subtypes. Conceivably, the effects induced by LHS may be mediated by different subtypes of opioid receptors depending on the side of the injury. The plastic response after right-sided LHS, which is characterized by right-hindlimb flexion, can involve δ-opioid receptors that are activated by their endogenous ligand, Leu-enkephalin. This peptide induces HL-PA with flexion of the right hindlimb, while the δ-opioid antagonist naltrindole blocks the effects of left-sided unilateral brain injury, producing a contralateral limb response [18,19,20]. Conversely, HL-PA with left flexion after a left-sided LHS may be mediated by endogenous dynorphins acting via κ-opioid receptors and antagonized by κ-opioid antagonists [18,19,20]. Determining whether targeting these receptor-defined signatures with subtype-selective antagonists can promote functional recovery or compensation of postural deficits could have translational value for conditions such as spinal cord injury, traumatic brain injury, and stroke.

### 4.3. Humoral Signaling in HL-PA Formation

An intriguing finding was that cervical LHS, performed after complete lumbar spinal transection, produced HL-PA with flexion of the contralesional hindlimb. In spinalized animals, signals from the hemisection site to hindlimb motoneurons may be transmitted through a humoral pathway (Figure 4). Specifically, the LHS-induced unilateral disruption of ascending sensory and proprioceptive signaling from spinal segments could provoke asymmetric responses in supraspinal structures, including the hypothalamic–pituitary system. This may trigger the release of side-specific signaling molecules into the bloodstream that then reach lumbar neurons or their projections onto hindlimb muscles, inducing postural asymmetry.

Previous studies revealed the topographic neuroendocrine system, which mediates the contralateral effects of unilateral brain lesions on the lumbar spinal cords in rats with completely transected thoracic or cervical spinal cords [17,20]. The humoral factors mediating the effects of left-sided injuries were identified as β-endorphin and Arg-vasopressin. In rats with intact brains, these peptides caused right-hindlimb flexion. In contrast, dynorphin and Met-enkephalin may transmit the effects of right-sided brain injuries and consistently cause left-hindlimb flexion when injected intrathecally into the caudal part of the transected spinal cord of rats with an intact brain [18,19,20].

Relevant to this study is that Met-enkephalin also induced HL-PA when it was administered intrathecally above the level of spinal cord transection [55]. However, in this experimental design, the hindlimb was flexed on the right side. Notably, the side of the response was reversed when the effects of both Met-enkephalin and LHS were transmitted through humoral pathways, compared to when they directly affected neurons in the lumbar spinal cord. In terms of its function, the reversal can counteract ipsilateral changes in reflexes and posture caused by hemisection-induced denervation of lumbar circuits. Naloxone reduced HL-PA only in the lumbar hemisection model and showed no significant effect after the cervical hemisection. Thus, the humoral transmission is not likely mediated by the opioid neurohormones.

One limitation of these findings is that they were obtained from anesthetized animals. The second is the role of the sympathetic nervous system. A previous study ruled out the sympathetic system as a left–right-side-specific signaling pathway from the injured brain to the lumbar spinal cord [20]. The hindlimb responses to unilateral brain injury were induced in rats with cervical spinal cord transections rostral to the preganglionic sympathetic neurons. In principle, the paravertebral chain of sympathetic ganglia—the only remaining neural connection after complete spinal cord transection—could convey signals from the cervical LHS to the hindlimb vasculature, affecting ipsilateral and contralateral muscles differently. However, several arguments do not support this mechanism. The sympathetic ganglia are likely not involved in control of lumbar neural circuits by the supraspinal structures [56,57]. Furthermore, sympathetic neurons have limited capacity for the selective regulation of blood flow to the left and right hindlimbs [58]. Nonetheless, the role of the sympathetic pathway in the effects of LHS on HL-PA remains to be elucidated.

## 5. Conclusions

This study demonstrates that spinal opioid receptor-mediated pathways and non-opioid humoral signaling both contribute to asymmetric postural deficits following spinal cord hemisection. The extent to which these neural and neuroendocrine side-specific mechanisms interact or compensate for each other—and how they regulate lateralized processes across distant CNS regions—remains an open question.

From a clinical standpoint, these findings highlight the potential of opioid receptor antagonists as therapeutic agents for alleviating side-specific neurological impairments after lateralized spinal injury. Moreover, identifying blood-borne factors that modulate or oppose side-specific opioid effects may inform novel strategies for restoring left–right motor balance in injury and disease.

## Figures and Tables

**Figure 1 cells-14-00980-f001:**
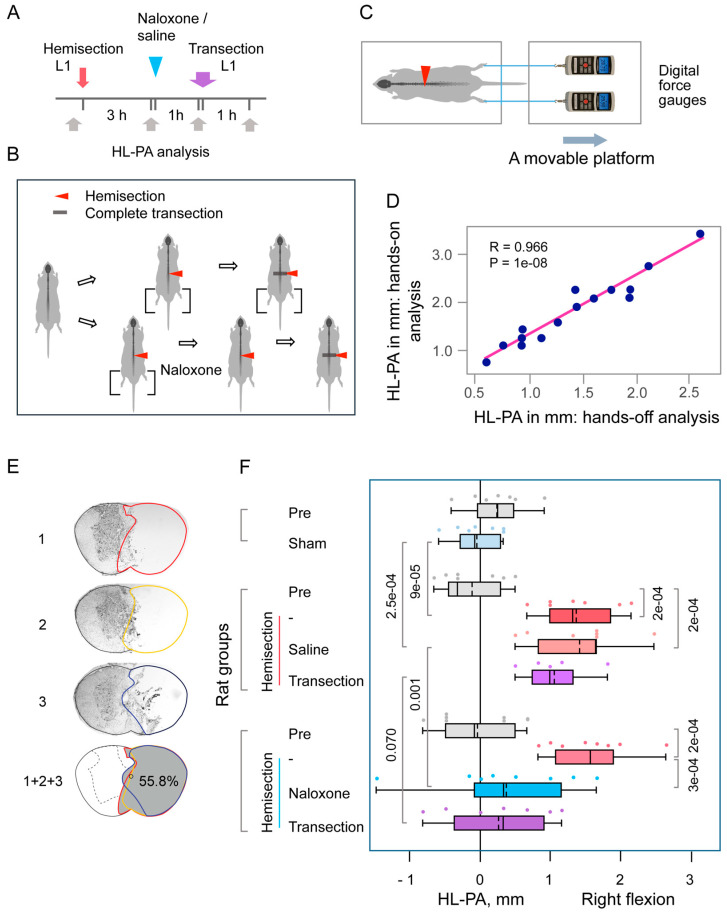
Hindlimb postural asymmetry (HL-PA) induced by right-side hemisection of the lumbar spinal cord and the effects of naloxone. (**A**,**B**) Experimental design: Hemisection was performed at the L1–L2 level, followed by administration of either saline (n = 7) or naloxone (n = 8) three hours later. One hour after injection, a complete spinal cord transection was performed by excising a 3–4 mm segment at the same level. Sham surgery was conducted as a control for hemisection in 7 rats. HL-PA was measured at four time points: before (Pre), three hours post-hemisection or sham surgery, one-hour post-naloxone or saline administration, and one-hour post-complete transection. (**C**) HL-PA was measured using a micromanipulator-controlled force meter consisting of two digital force gauges fixed on a movable platform. (**D**) Pearson correlation between the HL-PA size analyzed by the hands-off and hands-on assay (n = 15). Data were combined for the saline- and naloxone-treated rat groups, and they were analyzed 3 h after hemisection, before the administration of these solutions. (**E**) Representative images of the lesion site from a rat with a hemisection at the L1–L2 level. Images 1, 2, and 3 depict three adjacent sections representing the submaximal lesion area. The combined image (1 + 2 + 3) illustrates the maximal lesion area (shaded in gray), calculated by stacking the outlined regions from the three sections. In this rat, the lesion encompassed 55.8% of the spinal cord’s cross-sectional area, which included the right half and small part of the left half of the spinal cord. (**F**) HL-PA size in millimeters, with negative and positive values indicating flexion on the left and right sides, respectively. Boxplots show the distribution (minimum, first quartile, median, mean, third quartile, and maximum), with medians and means indicated by solid and dashed lines, respectively. Individual rat HL-PA values are shown by circles. Statistical analysis: Repeated-measures ANOVA revealed a main effect of repeated measurements (F(3,30) = 34.85, *p* < 1 × 10^−5^) and a significant interaction effect (F(3,30) = 6.87, *p* < 0.001) but no significant group effect (naloxone) (F(1,10) = 1.71, *p* < 0.22). Tukey’s HSD post hoc *p*-values are indicated on the plots. Two-tailed Student’s *t*-test compared hemisection and sham groups at the three-hour time point.

**Figure 2 cells-14-00980-f002:**
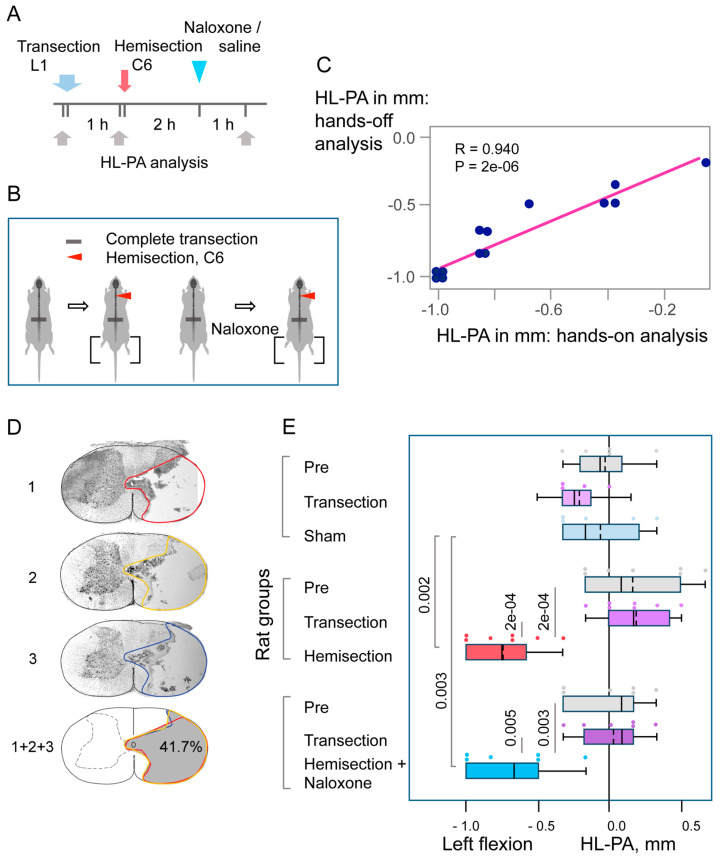
Hindlimb postural asymmetry (HL-PA) induced by right-side hemisection of the cervical spinal cord in rats with prior lumbar spinal transection and the effects of naloxone. (**A**,**B**) Experimental design: The lumbar spinal cord was transected completely at the L1–L2 level, followed by right-side hemisection at the C6–C7 level. Saline (n = 7) or naloxone (n = 6) was administered two hours post-hemisection. Five rats with transected spinal cords served as sham-operated controls. HL-PA was measured at three time points: before (Pre), one hour after complete transection, and three hours post-hemisection or sham surgery that was one-hour post-naloxone or saline injection. (**C**) Pearson correlation between the HL-PA size analyzed by the hands-off and hands-on assay. Data were combined for the saline- and naloxone-treated rat groups and were analyzed 3 h after hemisection, before the administration of these solutions (n = 13). (**D**) Representative images of the lesion site from a rat with a hemisection at the C6/7 level. Images 1, 2, and 3 depict three adjacent sections representing the submaximal lesion area. The combined image (1 + 2 + 3) illustrates the maximal lesion area (shaded in gray), calculated by stacking the outlined regions from the three sections. In this rat, the lesion encompassed 41.7% of the spinal cord’s cross-sectional area, predominantly affecting the right half. (**E**) HL-PA size in millimeters, with negative and positive values indicating flexion on the left and right sides, respectively. Boxplots show the distribution (minimum, first quartile, median, third quartile, and maximum), with medians and means indicated by solid and dashed lines, respectively. Individual rat HL-PA values are shown by circles. Statistical analysis: Repeated-measures ANOVA revealed significant effects of hemisection (F(2,32) = 21.79, *p* < 1 × 10^−5^) and interaction (F(4,32) = 7.28, *p* < 3 × 10^−4^), but no significant group effect (F(2,16) = 0.64, *p* < 0.54). Tukey’s HSD post hoc *p*-values are shown.

**Figure 3 cells-14-00980-f003:**
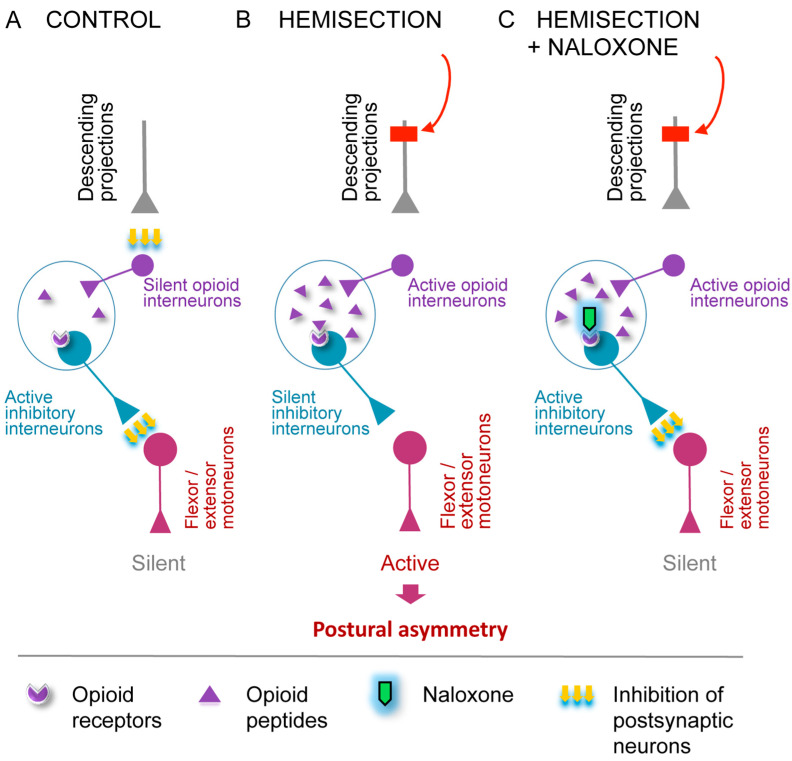
A model for the opioid receptors mediated ipsilateral effects of the LHS on hindlimb posture. (**A**) Descending projections provide tonic inhibition to opioid neurons in the spinal cord that regulate the activity of inhibitory interneurons projecting onto motoneurons. As a result, motoneurons, which innervate hindlimb extensors and/or flexors, remain inactive. (**B**) LHS abolishes tonic inhibition of spinal opioid neurons on the injury side, leading to (i) activation and release of opioid peptides by opioid interneurons; (ii) reduced activity of inhibitory interneurons projecting onto motoneurons; (iii) activation of motoneurons; and (iv) a resulting pathological response—HL-PA. (**C**) As in (**B**), spinal opioid interneurons are disinhibited on the ipsilateral side due to LHS. They release opioid peptides, for which interaction with opioid receptors expressed by inhibitory interneurons projection to motoneurons is blocked by the general opioid antagonist naloxone. Therefore, the inhibitory interneurons become active again and inhibit the motoneurons. This diminishes the pathological response on the ipsilesional side, reestablishing symmetry in posture and reflexes. The LHS-induced changes in opioid interneurons may account for the injury effects that persist after complete spinal cord transection.

**Figure 4 cells-14-00980-f004:**
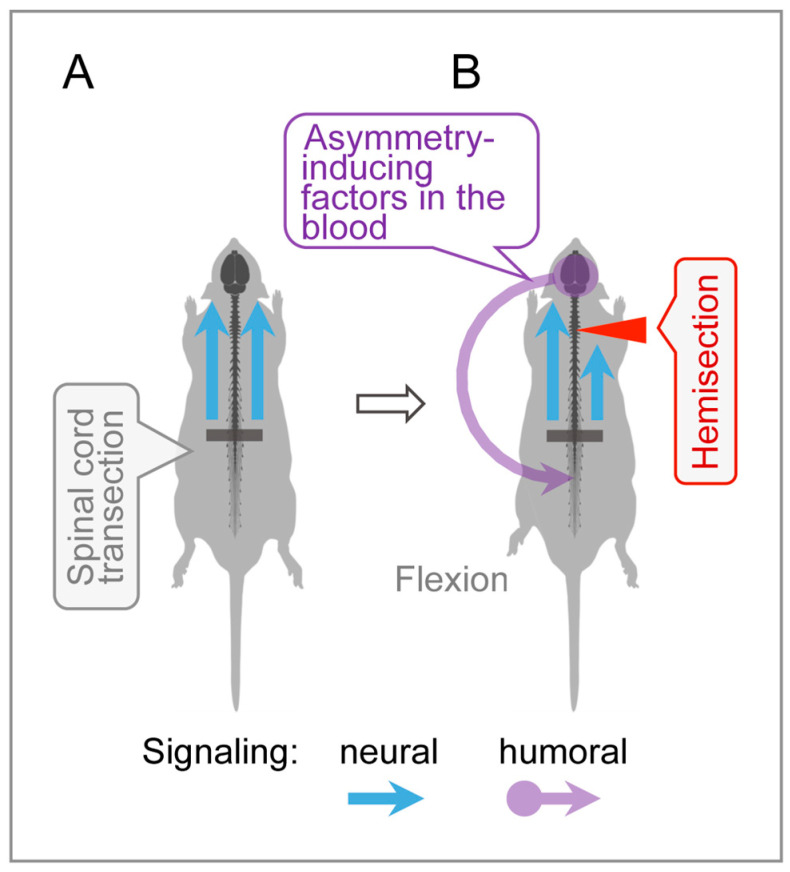
Humoral side-specific signaling from the cervical LHS site to the lumbar spinal cord. (**A**) Control animal with transected lumbar spinal cord. (**B**) In rats with transected lumbar spinal cords, LHS unilaterally disrupts ascending sensory and proprioceptive signals from spinal segments rostral to the lumbar transection, triggering asymmetric responses in supraspinal structures. This may lead to the release of asymmetry-inducing molecules, likely from the hypothalamic–pituitary system [17,20]. These molecules travel via the bloodstream to lumbar neurons or their projections onto hindlimb muscles, inducing contralesional hindlimb flexion.

## Data Availability

The data that support the findings of this study are available within the article and from the corresponding author upon reasonable request.

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
