# Peer review of "Acute Postural Effects of Spinal Cord Injury: Dual Neural Opioid and Endocrine Non-Opioid Mechanism"

_cells, 2025, doi:10.3390/cells14130980_

Round 1

Reviewer 1 Report

Comments and Suggestions for Authors

This is a well-written manuscript describing a series of studies examining the interesting phenomenon of postural asymmetry after a spinal cord hemisection, and that this asymmetry is dependent on endogenous opioids when the hemisection is at L1/2 but not when the hemisection is a C6/7. The studies are cleverly designed, use reasonable group sizes and appear to have been appropriately analyzed and interpreted. The primary findings are well laid-out and are of fairly significant interest because they further support the concepts of postural asymmetry, the important role of endogenous opioids and the involvement of extra-spinal, neuroendocrine mechanisms.

Concerns/critiques (quite minor).

In figure 1 you show naloxone preventing or reducing the asymmetry, but the variability for the naloxone and transection is very high. Please comment on this variability and what it might mean for the interpretation of the findings. 

In figure 3 the color used to represent inhibition of postsynaptic neurons is not as it appears in the legend, and this figure appears to be somewhat disordered. The cell bodies are not round, the text is covered by one of the cell bodies, and some text is too close to other text. This is an important figure for the paper and additional effort on it would be worthwhile.

It is stated that three rats per group had their lesion sites analyzed, but it doesn’t make it clear why this wasn’t done for all the animals?

There is a lot of methodological information contained in the figure legends rather than the results section, and this reviewer suggests that it be moved (e.g. order of events in the studies). However this is just a preference.

The discussion is longer than necessary and is not as concise as it could be. This somewhat detracts from the main messages which are very interesting and worthwhile.

Author Response

We are very grateful to the reviewer and editors for outstanding and encouraging analysis of our manuscript. We agree with many comments and suggestions, and have addressed them to the best of our ability. We respond to each of the points raised by the reviewers below.

Comments 1: “In figure 1 you show naloxone preventing or reducing the asymmetry, but the variability for the naloxone and transection is very high. Please comment on this variability and what it might mean for the interpretation of the findings.”

Response: We thank the reviewer for pointing this out. Indeed, the range (the difference between the maximum and minimum values) in the naloxone-treated animals is higher than it was before naloxone was administered to the same rats, as well as to the LHS rats treated with saline. However, a two-tailed, two-sample, unequal variance Student's t-test (assuming normal distribution of the data) demonstrated that the differences in HL-PA before and after naloxone administration in the same rat group and between the naloxone- and saline-treated rat groups are significant: P = 0.013 and 0.032, respectively (Page 7: Lines 272-279).

Furthermore, an F-test for equality of variances revealed no significant differences in standard deviations between time points before and after naloxone administration (SD1 = 0.7, SD2 = 1.0; F(7, 7) = 0.4, P = 0.201) or between the naloxone- and saline-treated groups (SD1 = 0.6, SD2 = 1.0; F(6, 7) = 0.5, P = 0.366).

The difference in range may be due to an outlier in the naloxone group that was not omitted from the statistical analysis. When the outlier was excluded, the standard deviation was nearly similar among the rat groups.

Comments 2: “In figure 3 the color used to represent inhibition of postsynaptic neurons is not as it appears in the legend, and this figure appears to be somewhat disordered. The cell bodies are not round, the text is covered by one of the cell bodies, and some text is too close to other text. This is an important figure for the paper and additional effort on it would be worthwhile”.

Response: We apologize for including a corrupted version of this figure in the manuscript. The figure has been modified according to the reviewer's request, and the new version is included in the resubmitted manuscript (Page 9).

Comments 3: “It is stated that three rats per group had their lesion sites analyzed, but it doesn’t make it clear why this wasn’t done for all the animals?”

Response: Thank you for highlighting this issue. Ideally, we would analyze the lesion sites for all the animals. However, this is impossible due to the design of the experiment. For rats with a hemisection at the L1/2 spinal cord segment, the spinal cord was transected four hours after hemisection; thus, the spinal cord from these segments could not be used for histology. Regarding rats with a hemisection at the L6/7 level, the rats were usually euthanized directly after a long-term experiment.

To assess the precision of the surgery, we performed a hemisection lesion in a separate group of rats, with three lesioned at the L1/2 level and three at the C6/7 level. After measuring hindlimb postural asymmetry, the rats were perfused and the spinal cord was used for histology. This is stated on page 3 of the revised manuscript (Lines 136-145).

Overall, there was little variation between the rats. Analysis of six spinal cords showed that the hemisection affected an average of 50% of the tissue on the right side at each hemisection level, with minimal variation between rats (mean = 49% and 50%, and SD = 5% at both levels, respectively).

Comments 4: “There is a lot of methodological information contained in the figure legends rather than the results section, and this reviewer suggests that it be moved (e.g. order of events in the studies). However this is just a preference.”

Response: Thank you for addressing this issue. We were indeed hesitant about where to place the detailed information on the experimental design. First, it is too early to include it in the Methods section. Second, the designs of the two experiments are complex and different from each other. Thus, it is necessary to describe the design in detail near the experimental data presented, i.e., in the figure legends. On the other hand, the design should be described in the Results section, but this section should not be overloaded with details. Therefore, we chose a compromise: to briefly describe the experimental design in the respective result sections and provide a complete description of the experiments in the figure legends (see please legends to Figures 1 and 2).

Comments 5: “The discussion is longer than necessary and is not as concise as it could be. This somewhat detracts from the main messages which are very interesting and worthwhile.”

Response: Thank you for the comment. The discussion section has been shortened, with the current version focusing primarily on the findings of the study (Pages 10-13).

Reviewer 2 Report

Comments and Suggestions for Authors

     This paper is an integrative study of neural and endocrine mechanisms regarding hindlimb postural asymmetry (HL-PA) after unilateral spinal cord amputation (LHS). It is of high scientific significance and has been validated by appropriate methods. In particular, the finding that both neural pathways via spinal opioid receptors and non-opioid humoral pathways are involved in HL-PA is highly novel and important in contributing to the understanding of the pathogenesis of spinal cord injury and central movement disorders. If the following points are appropriately addressed, this article is deemed acceptable for publication in this journal.

  1. Visual observation and photographic analysis are used to evaluate HL-PA, but information on the objectivity and reproducibility of the evaluation (e.g., inter-rater agreement) should be included.
  2. When discussing the effects of humoral pathways, a more detailed discussion of the involvement of the sympathetic nervous system should be included.
  3. The hypothetical model in Figure 3 is an important visual summary of the main points of this study. In particular, a clearer explanation of the change from panel B to C would enhance understanding.

     This study presents new findings on postural asymmetry after spinal cord injury and is a useful report that includes potential clinical applications. We hope that the paper will become even more complete by carefully addressing the points raised above.

Author Response

We are very grateful to the reviewer and editors for outstanding and encouraging analysis of our manuscript. We agree with many comments and suggestions, and have addressed them to the best of our ability. We respond to each of the points raised by the reviewers below.

Comments 1: Visual observation and photographic analysis are used to evaluate HL-PA, but information on the objectivity and reproducibility of the evaluation (e.g., inter-rater agreement) should be included.

Response: We thank the reviewer for this excellent comment. The methodology for HL-PA analysis has been extensively characterized and validated in our previous studies (Zhang et al., Brain Communications 2020; Watanabe et al., Brain Communications 2020; Bakalkin et al., Exp Brain Res 2021; Watanabe et al., Brain Communications 2021; Lukoyanov et al., eLife 2021; Watanabe et al., eNeuro 2021; Watanabe et al., FUNCTION 2024). These studies employed three complementary approaches: (i) hands-on visual assessment, (ii) hands-off visual analysis, and (iii) biomechanical evaluation of hindlimb musculo-articular resistance to stretch. The hands-off and biomechanical methods utilized a micromanipulator-controlled force measurement system equipped with two digital force gauges mounted on a movable platform. This setup enabled precise and unbiased quantification of hindlimb asymmetry (see Figure 1C). Validation of the method was conducted by co-authors H.W. and M.Z., who served as raters. Strong and statistically significant correlations among all three assessment methods confirmed the robustness and reproducibility of the HL-PA assay (see references [17,20]). Moreover, the presence of HL-PA correlated with differences in nociceptive withdrawal reflexes between the left and right hindlimbs, as measured by EMG following electrical stimulation in spinalized decerebrate rats (Zhang et al., Brain Communications 2020; Lukoyanov et al., eLife 2021).

Importantly, similar HL-PA data were obtained across experimental paradigms investigating neurotrauma- and opioid-induced effects, with replication across three laboratories: Uppsala University, Lund University, and the University of Porto.

In the current study, both hands-on and hands-off assessments were applied, yielding highly correlated measurements of HL-PA magnitude (Figures 1 and 2). All reported data in the manuscript derive from the unbiased hands-off method. Additionally, data processing and statistical analyses were conducted by an independent statistician blinded to the experimental conditions.

These details, including the correlation data and validation approach, have been incorporated into the Methods, Results, and Figures 1 and 2 of the revised manuscript (e.g., see Page 4: Lines 153-183; Page 5: lines 205-217).

Comments 2: When discussing the effects of humoral pathways, a more detailed discussion of the involvement of the sympathetic nervous system should be included.

Response: A previous study ruled out the sympathetic system as a left-right side-specific signalling pathway from the injured brain to the lumbar spinal cord [20]. The hindlimb responses to unilateral brain injury were induced in rats with cervical spinal cord transections rostral to the preganglionic sympathetic neurons. However, a role for the sympathetic pathway in the effects of LHS on HL-PA remains to be elucidated.

In principle, the paravertebral chain of sympathetic ganglia — the only remaining neural connection after complete spinal cord transection — could convey signals from the cervical LHS to the hindlimb vasculature, affecting ipsilateral and contralateral muscles differently. However, several arguments do not support this mechanism. The sympathetic ganglia are likely not involved in control of lumbar neural circuits by the supraspinal structures. Furthermore, sympathetic neurons have limited capacity for the selective regulation of blood flow to the left and right hindlimbs.

This discussion has been added to the manuscript (Page 13: Lines 509-522).

Comments 3: The hypothetical model in Figure 3 is an important visual summary of the main points of this study. In particular, a clearer explanation of the change from panel B to C would enhance understanding.

Response: Once again, we apologize for including an early version of this figure in the manuscript. The figure has been modified according to the reviewer's request, and the updated version is included in the resubmitted manuscript. Specifically, we have provided a clearer explanation of the effects of naloxone depicted in C (Pages 9 and 10: Lines 334-347).

Reviewer 3 Report

Comments and Suggestions for Authors

The phenotypic effects reported in this study are highly interesting and provide valuable insights into postural asymmetry following lateral spinal cord injury. However, the underlying mechanisms are not thoroughly investigated. Although the effect of naloxone suggests the involvement of the opioid system, no additional mechanistic data are presented. To enhance the impact of the study, it would be important to complement the current findings with data that further clarify the molecular or cellular mechanisms. In addition, the discussion leans heavily on existing literature and does not sufficiently elaborate on the novel findings of this work. For instance, the potential involvement of specific opioid receptor subtypes is not addressed and should be considered to deepen the mechanistic understanding.

In particular, I believe that actual experimental data should be presented instead of a conceptual diagram, in order to more effectively support the authors’ claims.

Author Response

We are very grateful to the reviewer and editors for outstanding and encouraging analysis of our manuscript. We agree with many comments and suggestions, and have addressed them to the best of our ability. We respond to each of the points raised by the reviewers below.

Comments 1: it would be important to complement the current findings with data that further clarify the molecular or cellular mechanisms.

Response: We agree and thank the reviewer for this encouraging comment. However, before such an analysis is initiated, the LHS model should be characterised in more detail at system, physiological and biochemical levels. It should therefore be examined whether the effects of the left-side LHS can also be transmitted via side-specific neurohormonal messages and whether the left- and right-side messages are mediated by neuropeptide/neurohormonal receptors. The receptor type and subtype should also be revealed and the nature of the hormonal signals mediating the effects of the left and right LHS should be identified biochemically. Our subsequent plan is to focus on identifying DEGs affected by the left and right LHS, as well as gene co-expression networks in the hypothalamus involved in left-right, side-specific signalling of lateralised SCI, as previously described (Watanabe et al., 2024).

Comments 2: In addition, the discussion leans heavily on existing literature and does not sufficiently elaborate on the novel findings of this work. For instance, the potential involvement of specific opioid receptor subtypes is not addressed and should be considered to deepen the mechanistic understanding.

Response: With all due respect, we can only partially agree with this comment. In the revised version, however, we focus the discussion primarily on our findings, whereas discussion of published data has been substantially reduced or eliminated. Furthermore, the entire discussion section is devoted to the role of the opioid system in the mechanisms of the LHS. While the discussion of opioid receptor subtypes may be premature for analysing SCI mechanisms, we have nevertheless included it in the manuscript (Pages 11 and 12: Lines 398-480).

Comments 3: In particular, I believe that actual experimental data should be presented instead of a conceptual diagram, in order to more effectively support the authors’ claims.

Response: This is the first phenomenological study to examine the opioid and endocrine mechanisms underlying the effects of lateralised SCI.

The set of experimental data presented, the statistical significance of the effects, and the proven efficiency of the methodological approach in identifying lateralised, side-specific neurotrauma effects (see, for example, Zhang et al., Brain Communications, 2020; Watanabe et al., Brain Communications, 2020 and 2021; FUNCTION, 2024; eNeuro, 2021; Lukoyanov et al., eLife, 2021) all demonstrate the robustness of this spinal phenomenon.

Postural asymmetry is a well-reproduced, clinically relevant model. Taking these factors into account, we believe that our claims are well supported by this dataset. At the same time, the conceptual diagrams presented in Figures 3 and 4 are necessary to explain the emerging concept, which is unfamiliar to general neuroscientists and neurotrauma experts.  

Round 2

Reviewer 3 Report

Comments and Suggestions for Authors

Although the Discussion section has been revised, no additional molecular or cellular data—such as those expected within the scope of Cells—have been included. Therefore, I do not consider the manuscript suitable for acceptance in its current form.

Author Response

Comment 1: Although the Discussion section has been revised, no additional molecular or cellular data—such as those expected within the scope of Cells—have been included. Therefore, I do not consider the manuscript suitable for acceptance in its current form.

Response: We would like to thank the reviewer for taking the time to analyse our manuscript. However, with all due respect, we disagree with the latest statement due to the following formal and scientific issues.

Formal issues:

The Scope of “Cells covers every topic related to cell biology and PHYSIOLOGY, molecular biology, and biophysics.” Examples of physiological topics include the following 'CELLS' topics: 'Pain perception and its management', 'Brain-machine interfacing and neuroprosthesis', and 'Nerve-electronics conduits'. We would not expect these topics to include molecular or cellular data. Therefore, it appears that studies with only physiological and pharmacological results and no molecular data could be published in the journal.

Scientific issues:

Figure 3 presents a cellular synaptic model that aligns with the scope of the journal and is therefore sufficient to justify publication of this manuscript in 'Cells'.

As we have already emphasised, the in-depth molecular / cellular mechanistic analysis requested is premature at this stage of project development. This is because the LHS model requires further characterisation at system and physiological levels. We should examine whether the effects of the left-side LHS can be transmitted via side-specific neurohormonal messages, and whether left- and right-side messages are mediated by neuropeptide/neurohormonal receptors. The receptor type and subtype should also be determined, as should the nature of the hormonal signals that mediate the effects of the left and right LHS.

Simply adding data on gene expression, peptides or proteins, or immunohistochemistry (IHC) data would not significantly improve our understanding of the phenomenon discovered. Understanding it in molecular, biochemical and cellular terms is a challenging project that is definitely beyond the scope of this study.